# Improving the quality of routine maternal and newborn data captured in primary health facilities in Gombe State, Northeastern Nigeria: a before-and-after study

Antoinette Alas Bhattacharya  ,[1] Elizabeth Allen,[2] Nasir Umar,[1] Ahmed Audu,[3] Habila Felix,[3] Joanna Schellenberg,[1] Tanya Marchant[1]

► Prepublication history and additional materials for this paper is available online. To view these files, please visit the journal online (http://dx.doi.org/10.1136/bmjopen-2020-038174).

¹Department of Disease Control, London School of Hygiene & Tropical Medicine Faculty of Infectious and Tropical Diseases, London, UK
²Department of Medical Statistics, London School of Hygiene & Tropical Medicine Faculty of Epidemiology and Population Health, London, UK
³Gombe State Primary Health Care Development Agency, Gombe, Nigeria

**Correspondence to**
Antoinette Alas Bhattacharya;
antoinette.bhattacharya@lshtm.ac.uk

## ABSTRACT

**Objectives** Primary objective: to assess nine data quality metrics for 14 maternal and newborn health data elements, following implementation of an integrated, district-focused data quality intervention. Secondary objective: to consider whether assessing the data quality metrics beyond completeness and accuracy of facility reporting offered new insight into reviewing routine data quality.

**Design** Before-and-after study design.

**Setting** Primary health facilities in Gombe State, Northeastern Nigeria.

**Participants** Monitoring and evaluation officers and maternal, newborn and child health coordinators for state-level and all 11 local government areas (district-equivalent) overseeing 492 primary care facilities offering maternal and newborn care services.

**Intervention** Between April 2017 and December 2018, we implemented an integrated data quality intervention which included: introduction of job aids and regular self-assessment of data quality, peer-review and feedback, learning workshops, work planning for improvement, and ongoing support through social media.

**Outcome measures** 9 metrics for the data quality dimensions of completeness and timeliness, internal consistency of reported data, and external consistency.

**Results** The data quality intervention was associated with improvements in seven of nine data quality metrics assessed including availability and timeliness of reporting, completeness of data elements, accuracy of facility reporting, consistency between related data elements, and frequency of outliers reported. Improvement differed by data element type, with content of care and commodity-related data improving more than contact-related data. Increases in the consistency between related data elements demonstrated improved internal consistency within and across facility documentation.

**Conclusions** An integrated district-focused data quality intervention—including regular self-assessment of data quality, peer-review and feedback, learning workshops, work planning for improvement, and ongoing support through social media—can increase the completeness, accuracy and internal consistency of facility-based routine data.

### Strengths and limitations of this study

► We extended the evidence on integrating data quality interventions within existing systems to improve the quality of facility-based data for monitoring and planning.

► We demonstrated the value of an integrated district-focused data quality intervention to include regular self-assessments of data quality, peer-review and feedback, work planning for improvement, and ongoing support through social media.

► We assessed the usefulness of the WHO's catalogue of data quality metrics to measure and monitor the quality of routine facility data, as data quality studies primarily review completeness and accuracy of facility reporting.

► Without a concurrent comparison group, our before-and-after analyses cannot eliminate the effects of concurrent events and activities on data quality metrics.

## INTRODUCTION

Routine health information systems provide essential data for governments and stakeholders to make decisions for managing performance and optimising service delivery.[1–3] Routine health information systems, which include facility-based data, have the potential to provide disaggregated statistics important for understanding disparities and inequities in the provision of quality services and related health outcomes.[4 5]

Effective use of routine health data is dependent, in part, on the quality of data.[1 4 6–9] Studies assessing the quality of routine health data have shown persistent challenges in incomplete and untimely reporting, incomplete indicator-level data, inaccurate facility reporting, and imprecise target population estimates for coverage.[10–13] Further, studies

have noted considerably poorer data quality at the facility-level than at district, state and national-levels, citing challenges in accurately capturing data as well as in tallying and summarising service data for monthly reporting.[14–16]

Improving the quality of routine data is a priority given the potential of routine data to contribute to effective programme monitoring and performance management. Efforts to improve the quality of routine data have included data quality checking with feedback as a low-resource and low-cost activity for strengthening data quality, particularly when implemented as a routine activity. Data quality checking with feedback have been shown to improve completeness, timeliness and accuracy of facility reporting.[4 6 7 10 17 18] Knowledge transfer activities, such as mentoring, training or workshops, have offered an opportunity to receive additional skills building in data quality checking or data use.[6 8 19–28] Knowledge sharing activities such as data review meetings and dashboards have brought together different health system-levels or peers within a health system level and include understanding the relative performance of teams on service coverage indicators or on dimensions of data quality.[6 8 20 22 29–31] The introduction of technology, either software such as District Health Information Software V.2 (DHIS2) or a device to enhance data collection such as a tablet, has demonstrated improvements in completeness, timeliness and error detection. However, these technology-based initiatives often required supervision, monitoring and feedback to ensure errors were resolved and that data entered were accurate and consistent.[7 27 32 33] Activities which aligned with user priorities and were integrated within existing government systems were perceived to be advantageous as well as more likely to be adopted and adapted.[20]

Within the context of routine health information systems, the WHO has characterised routine data quality into four broad dimensions: completeness and timeliness; internal consistency; external consistency and external comparisons.[34] While the above-mentioned data quality interventions have demonstrated increases in completeness and accuracy of facility reporting,[4 6 8 35 36] there are few peer-reviewed studies that quantitatively assessed changes in data quality metrics beyond this.[3 9 37]

In this study, our primary objective was to measure the changes in data quality metrics before and after the introduction of an integrated district-focused intervention in Northeastern Nigeria for routine facility data captured in primary health facilities. A secondary objective was to determine the extent to which expanding data quality metrics beyond completeness and accuracy of facility reporting offered new insight into reviewing data quality.

## METHODS
### Study design
This was a before-and-after study design for a data quality intervention in all 11 local government areas (LGA, district-equivalent) of Gombe State, Northeastern Nigeria. We present results for the state (n=492 facilities), comparing the 21-month period before the intervention, July 2015–March 2017, with the 21-month period after introducing the intervention, April 2017–December 2018.

### Patient and public involvement
Patients or the public were not involved in the design, conduct, reporting or dissemination of the research described here.

### Study setting
Gombe State is located in Northeastern Nigeria, a region with high maternal and newborn mortality at 1549 per 100 000 live births and 35 per 1000 live births, respectively.[38 39] With an estimated population of 2.9 million, Gombe is predominantly rural and 35% of the women have some primary school education.[40 41] Most women access maternity care through public facilities. Seventy-two per cent of women reported at least one antenatal care visit during their last pregnancy and 28% gave birth in a health facility.[39 41] In 2018, over 70% of facility deliveries took place in rural primary health facilities.[42]

Under Nigeria's national policy of *Primary Health Care Under One Roof*, the Gombe State Primary Health Care Development Agency oversees the administration and service delivery for primary health facilities across 11 LGA; each LGA has 10–11 political wards (114 wards, total).[43 44] LGA monitoring and evaluation officers are responsible for community-level and facility-level data collection, validation and reporting to the state office. LGA maternal, neonatal and child health coordinators support the supervision and implementation of services for women and children.

During the intervention period, Gombe State had 492 primary health facilities providing antenatal and childbirth services. As in other states in Nigeria, Gombe facility staff generally completed 13 paper-based registers to document the services they provide (Nigeria health management information system, V.2013). Every month, a subset of data in these registers were tallied and summarised in a paper-based report and sent to the LGA health office to be entered into DHIS2.

### Data quality intervention
The routine data quality intervention period spanned 21 months, from April 2017 through December 2018. Following a situational analysis and literature review, development of the intervention took place in consultation with the Gombe State Primary Health Care Development Agency. The design was predicated on three key issues. Interventions that include data quality assessments and feedback have demonstrated improvement in completeness, timeliness and consistency.[4 6 8 19 20 22 23 26–28] Monitoring and evaluation officers and programme coordinators in the state were expected to engage with data from facilities to support improvement of service delivery; however, their interactions were not optimally structured

to ensure efficient and complementary engagement with facility data. Further, the recent publication of the WHO data quality review toolkit in early 2017 provided clarity on assessing the data quality dimensions and metrics for facility data; however, the Gombe district staff required further support and job aids to administer the data quality reviews as recommended in the WHO guidance.[45]

Thus, the intervention emphasised the partnership between the LGA monitoring and evaluation officer and the LGA maternal, newborn and child health programme coordinator to underscore the link between the quality and use of routine data.[8] The intervention was designed to facilitate existing LGA-level supervision responsibilities. It leveraged scheduling of ongoing activities to minimise cost, added job aids based on the WHO data quality review toolkit, and defined performance standards to provide structure to existing data quality checking duties and to target feedback to facilities.

The intervention included the following activities: (i) data quality learning workshops to present data quality self-assessment findings and develop work plans for improvement; (ii) defining data quality performance standards and milestones for completeness, timeliness and consistency; (iii) introduction of job aids to self-assess data quality according to the WHO data quality metrics; (iv) monthly state-level and LGA-level data quality summary reports; (v) intentional practice on providing constructive feedback to peers and low-performing facilities to promote a positive culture of data use and (vi) ongoing engagement on data quality issues through government-approved communication channels, including the social media application WhatsApp.

Twenty-six main attendees participated in the workshops and ongoing communication in-between workshops. These included two participants from each of the 11 LGAs, a monitoring and evaluation officer and the maternal, newborn and child health programme coordinator. At the state-level, four officials participated: the director of the Gombe State Planning, Research, and Statistics Department, the state monitoring and evaluation officer, the state health management information systems officer, and the state maternal, newborn and child health coordinator.

Four data quality learning workshops took place every 6–9 months. The 2-day workshops included the introduction of job aids and practical sessions to strengthen data quality checking skills, the presentation of the state's and each LGA's self-assessment of data quality, and the development of 6-month work plans to improve the quality of facility-based routine data. Using materials designed for postgraduate learning and teaching, there was intentional practice on how to provide constructive feedback to peers and facilities to promote a positive culture of information use.[46]

At each workshop, a major theme emerged during the work planning sessions (figure 1). For example, at the first workshop, participants were concerned with inconsistencies observed between the paper-based facility

| Workshop 1 | Workshop 2 | Workshop 3 | Workshop 4 |
|---|---|---|---|
| *Learning and practice:* <br>• Data quality dimensions, metrics <br>• Job aids for self-assessment <br>• Work planning for improvement <br><br>*Participant presentation:* <br>Present findings for select data quality metrics (completeness of reporting and agreement between facility register data and reports) <br><br>*Main outputs from work planning:* <br>• Finalise standards and milestones for data quality reporting by LGAs <br>• Revitalise monthly LGA validation committee <br>• Revitalise WhatsApp group for ongoing communication | *Learning and practice:* <br>• Quarterly self-assessment practice with job aids <br>• Interpreting and visualising self-assessment findings <br><br>*Participant presentation:* <br>Present quarterly self-assessment findings for peer review and feedback <br><br>*Main outputs from work planning:* <br>• Adjust data quality metric calculations to exclude inactive facilities and indicators in DHIS2 <br>• Improve completeness of data elements <br>• Identify facilities with low data quality metrics and call/visit them to problem-solve | *Preparation before workshop:* <br>• Quarterly data quality self-assessment, January-March 2018 <br>• Investigate reasons for facilities' higher/lower data quality metrics <br><br>*Learning and Practice:* <br>• Interpreting and visualising self-assessment findings <br>• Comparing LGA-level findings with performance of Gombe State and with neighbouring LGA <br>• Providing positive and constructive feedback <br><br>*Participant presentation:* <br>Joint presentation (with neighbouring LGA) of quarterly review findings <br><br>*Main outputs from work planning:* <br>• Clean up inactive facilities and indicators in DHIS2 <br>• Identify facilities with low data quality metrics and call/visit them to problem-solve <br>• Practice positive feedback | *Preparation before workshop:* <br>• Bi-annual data quality self-assessment, May-October 2018 <br>• Investigate reasons for facilities' higher/lower data quality metrics <br><br>*Learning and Practice:* <br>• Elements of an effective presentation and feedback <br><br>*Participant presentation:* <br>• Bi-annual data quality findings and priorities for next 6 months <br>• Demonstrating positive feedback to presenters <br><br>*Main outputs from work planning:* <br>• Improve consistency between related data elements, focus feedback on facilities with higher inconsistency between first antenatal care visits and related services <br>• Provide faster, real-time feedback to facilities |

**Figure 1** Data quality learning workshops in Gombe State, April 2017–December 2018. DHIS2, District Health Information Software V.2; LGA, local government areas.

registers and the facility's monthly summary reports. Activities enacted from the work planning session were to revitalise dormant strategies previously set up to address programme monitoring and evaluation activities: (i) LGA data validation committee meetings, where facilities bring their registers for verification against their submitted monthly facility report and (ii) a social media WhatsApp group of LGA actors and facilities. The LGA teams posted pictures and comments on these facility interactions on the WhatsApp group for encouragement and accountability.

After the first workshop, the Gombe State monitoring and evaluation officer disseminated monthly state-level and LGA-level data quality summary reports. LGAs were assessed according to the WHO data quality metrics and recommendations for improvement were offered. Initially, this activity was designed for the external workshop facilitators to compose and disseminate, while building the capacity of the state officer to take on this task over time.

### Outcomes

Using the WHO data quality review toolkit for routine facility data, we assessed nine metrics across the three data quality dimensions of completeness and timeliness; internal consistency; and external consistency.[45] The data sources and analyses for each data quality metric are described in the following section. Online supplemental table S1 provides additional information on each data quality metric assessed and the data sources reviewed.

### Data analysis and data sources

Three data sources were used to assess the routine data quality metrics, described later: facility-reported data in DHIS2, external facility surveys, and external household surveys.

DHIS2 contained monthly reports for the 492 primary facilities providing antenatal care, childbirth, and postnatal care services. Monthly aggregated DHIS2 data for July 2015–December 2018 were downloaded at one time and included 14 maternal and newborn health-related data elements.

These data were used to assess availability of facility reporting; timeliness of facility reporting; completeness of all 14 priority maternal and newborn health data elements, per monthly facility report; completeness of data element; presence of moderate and extreme outliers; consistency of indicator values over time and consistency between related data elements.

To assess completeness of facility reporting, we compared the proportions of facilities submitting reports before the intervention and during the intervention. To assess timeliness of facility reporting, the numerator for completeness of facility reporting was further limited to facilities that submitted the reports by the given timelines. To assess both completeness of data-related metrics, we compared the proportions of facility reports where a value was present for the relevant data. To compare the

proportions of the preintervention and intervention periods, we used linear mixed models to account for the clustering of facility measurements within facilities and the clustering of facilities within districts. P values for these metrics were computed from the Wald test within the mixed models. In these models, we also adjusted for potential confounders such as total client volume and time.[4 6]

We calculated the intraclass correlation coefficient (ICC) as a measure of agreement for the WHO metric assessing consistency between related data elements. ICC values range between 0 and 1, where values approaching 1 represented greater agreement between two related data elements. As the intervention worked through district-level staff to improve data quality across 492 primary health facilities, a paired t-test was carried out to compare preintervention and intervention periods matching ICCs at the district-level.

In 2016 and 2018, facility-level surveys were conducted in 97 primary facilities across Gombe to assess their readiness to provide maternal and newborn health services. The two surveys represented the approximate midpoints of the preintervention and intervention period. The selected facilities were a state-wide random sample drawn from all primary health facilities. Detailed methods are reported elsewhere.[47] The facility survey protocol was similar to a Service Availability and Readiness Assessment and also included data extraction from the facility's paper-based antenatal and postnatal care register and the labour and delivery register (Nigeria health management information system, V.2013).[48] A trained third party data collection team tallied and recorded the register data for each month of the 6-month periods immediately prior to the survey: January–June 2016 and February–July 2018.

These data were used to assess the accuracy of facility reporting (also referred to as data accuracy, data verification or concordance in peer-reviewed literature). We compared the facilities' paper-based registers data with the facilities' monthly reported data in DHIS2. As with the consistency between related data elements metric, we calculated the ICC as a measure of agreement and carried out a paired t-test for the preintervention and intervention periods matching ICCs at the district level. ICC values approaching 1 represented greater agreement.

In 2016 and 2018, household-level surveys were conducted in the enumeration areas of the abovementioned 97 primary facilities to assess access to and quality of maternal and newborn services.[47] A total of 79 enumeration areas were surveyed since some enumeration areas were served by more than one facility. All households in each enumeration area were surveyed. The household surveys included a mother's module which asked all women who reported a birth in the last year a detailed set of questions about their contact with health services across the continuum of care from pregnancy to postnatal care. Informed consent was obtained at the community leadership-level and at the individual-level

for each respondent. All invited participants agreed to be interviewed.

These data were used for external consistency during the preintervention and intervention periods. We compared coverage estimates from household surveys to those from the 97 matching facilities in DHIS2. We compared the same recall period for the surveys and the DHIS2. The DHIS2 data are considered consistent if they fall within the CIs of the external household survey estimates. Calculations of point estimates and their 95% CIs were done using the svyset Stata 14.2 command to adjust for clustering. We chose the highest-order clustering level to provide the most conservative CI estimates.[49]

## RESULTS

An integrated district-focused data quality intervention was implemented across 11 LGAs overseeing 492 primary

health facilities providing maternal and newborn care services. Below, we present the results for nine data quality metrics.

### Completeness and timeliness

Table 1 summarises the completeness and timeliness of reporting at the facility-level and indicator-level. At the facility-level, the availability of monthly facility reports improved from 72% to 82% (p<0.001) and timeliness of submitting the reports increased from 60% to 72% (p<0.001). The proportion of facility-months where all 14 priority maternal and newborn health data elements contained a value within the monthly report increased from 62% to 68% (p<0.001).

At the indicator-level, 7 of 14 data elements assessed improved in completeness compared with the preintervention period. Indicator-level completeness did not

| Table 1 Facility-level and indicator-level completeness and timeliness, Gombe State (n=492 facilities) | | |
|---|---|---|
| | **Preintervention** | **Intervention** |
| | July 2015–March 2017 | April 2017–December 2018 |
| | % (95% CI) | % (95% CI) |
| **Facility-level** | | |
| Availability of monthly facility reports | 72 (69 to 74) | 82 (80 to 84) |
| Timeliness of monthly facility reports | 60 (57 to 62) | 72 (70 to 74) |
| Completeness of all 14 priority maternal and newborn health data elements, per monthly facility report | 62 (60 to 63) | 68 (66 to 70) |
| **Indicator-level** | | |
| For every 100 facilities that submitted a monthly facility report, the percentage of facilities reporting a value for the following services | | |
| First antenatal care visits | 76 (67 to 85) | 77 (70 to 84) |
| Total antenatal care visits | 100 (99 to 100) | 100 (100 to 100) |
| Facility deliveries | 68 (59 to 77) | 67 (60 to 74) |
| For every 100 facilities that reported a value for first antenatal care visit, the percentage of facilities reporting a value for the following services | | |
| Antenatal care anaemia testing | 28 (16 to 39) | 36 (24 to 49) |
| Antenatal care syphilis testing | 42 (23 to 61) | 29 (23 to 35)* |
| Iron-folic acid supplementation | 80 (75 to 84) | 89 (85 to 92) |
| At least one dose administered of intermittent preventive treatment of malaria | 45 (34 to 56) | 56 (49 to 62) |
| At least one dose administered of tetanus toxoid vaccine | 90 (86 to 93) | 89 (86 to 91) |
| For every 100 facilities that reported a value for a facility delivery, the percentage of facilities reporting a value for the following services | | |
| Delivery by skilled birth attendant | 43 (25 to 61) | 86 (81 to 91) |
| Live birth or stillbirth | 90 (86 to 95) | 96 (94 to 97) |
| Baby weighed at birth | 89 (83 to 95) | 95 (94 to 97) |
| Oral polio vaccine given at birth | 79 (70 to 87) | 86 (82 to 90) |
| Early postpartum-postnatal care within 3 days of birth | 45 (38 to 53) | 55 (46 to 64) |
| BCG vaccine given during postnatal care period | 79 (71 to 88) | 81 (77 to 86) |

*During the intervention period, commodities for antenatal care syphilis testing were redistributed and restricted to 57 facilities. For these 57 facilities, completeness of data for antenatal care syphilis testing increased from 48% (95% CI 28 to 68) to 77% (95% CI 69 to 86).

**Table 2** Internal consistency: consistency between data elements with a predictable relationship (n=492 facilities)

| Data elements with a predictable relationship | ICC, Preintervention (95% CI) | ICC, Intervention 95% CI | P value |
|---|---|---|---|
| *Concurrent tallying within and across facility documentation* | | | |
| Normal+caesarean+assisted deliveries=livebirths+stillbirths | 0.83 (0.73 to 0.94) | 0.95 (0.93 to 0.98) | 0.023 |
| Total postnatal care visits=sum (postnatal care visit categories) | 0.46 (0.36 to 0.57) | 0.76 (0.66 to 0.85) | <0.001 |
| *Relationship between first antenatal care visits and* | | | |
| Antenatal care anaemia testing | 0.39 (0.18 to 0.59) | 0.51 (0.35 to 0.68) | 0.016 |
| Antenatal care syphilis testing | 0.19 (0.09 to 0.29) | 0.49 (0.37 to 0.60) | 0.003 |
| At least one dose of intermittent preventive treatment of malaria | 0.33 (0.17 to 0.48) | 0.43 (0.32 to 0.54) | 0.129 |
| At least one dose of tetanus-toxoid vaccine | 0.67 (0.53 to 0.80) | 0.73 (0.65 to 0.82) | 0.167 |
| Relationship between total antenatal care visits and | | | |
| Iron-folic acid supplementation | 0.92 (0.89 to 0.95) | 0.97 (0.95 to 0.99) | 0.005 |
| Relationship between facility deliveries and | | | |
| Baby weighed at birth | 0.82 (0.71 to 0.93) | 0.94 (0.92 to 0.97) | <0.001 |
| Delivery by skilled birth attendant | 0.42 (0.26 to 0.59) | 0.87 (0.80 to 0.94) | <0.001 |
| Oral polio vaccine given at birth | 0.48 (0.32 to 0.63) | 0.72 (0.65 to 0.79) | 0.007 |
| Early postpartum-postnatal care within 3 days of birth | 0.09 (0.06 to 0.13) | 0.19 (0.13 to 0.24) | <0.001 |
| BCG vaccine given during postnatal period | 0.56 (0.40 to 0.72) | 0.67 (0.57 to 0.76) | 0.147 |

ICC values range from 0 to 1, with values approaching 1 representing greater agreement.
ICC, intraclass correlation coefficient.

change for contact indicators such as first antenatal care visits, total antenatal care visits and facility deliveries.

### Internal consistency: consistency between related data elements

To assess the consistency between related data elements with a predictable relationship, two types of relationships were reviewed (table 2). The first type of relationship assessed concurrent tallying across different data elements within and across facility registers. For example (i) normal deliveries+caesarean deliveries+assisted deliveries=live births+still births and (ii) total postpartum visits reported=sum of the postnatal visit categories reported. For Gombe State, the ICC of delivery types (normal, caesarean, assisted) to birth types (live births, still births) improved from 0.83 (95% CI 0.73 to 0.94) to 0.95 (95% CI 0.93 to 0.97). Similar patterns of improvement were noted for postnatal visit tallying from an ICC of 0.46 (95% CI 0.36 to 0.57) to 0.76 (95% CI 0.66 to 0.85).

The second type of relationship assessed was a service provision compared with a contact indicator (eg, the number of antenatal care syphilis testing done compared with antenatal care first visits, the number of babies weighed at birth compared with the number of facility deliveries). During the preintervention period, 1 of the 10 relationships reflected high consistency: iron-folic acid supplementation. During the implementation period, 7 of the 10 relationships reflected improved consistency.

### Internal consistency: accuracy of facility reporting

Comparing the facilities' registers with their submitted monthly reports, accuracy of facility reporting (data accuracy) had improved for 6 of 7 indicators, reflecting greater agreement during the intervention period (table 3). For total antenatal care visits, there was considerable variation between districts during the preintervention and intervention periods, with an ICC of 0.62 (95% CI 0.41 to 0.83) and 0.86 (95% CI 0.72 to 0.99), respectively.

### Internal consistency: outliers and consistency over time

Online supplemental tables S2 and S3 summarise the presence of outliers and the consistency over time for the 14 maternal and newborn data elements. The frequency of months when outliers were reported decreased during the intervention period. However, 11 moderate outliers were reported during the intervention period compared with eight moderate outliers and two extreme outliers reported during the preintervention period. All 11 outliers reported during the intervention period occurred in May 2018 during a health worker strike. Of the 14 data elements assessed, 6 data elements were inconsistent over time due to reported increases in services when comparing the final year of the intervention 2018 to the mean value of the last 3 years and when comparing the preintervention and intervention periods.

**Table 3** Accuracy of facility reporting: comparison of paper-based facility records and facility monthly reports in District Health Information Software 2 (n=97 facilities)

| Data element | ICC, Preintervention (95% CI) | ICC, Intervention (95% CI) | P value |
|---|---|---|---|
| First antenatal care visits | 0.57 (0.40 to 0.75) | 0.88 (0.83 to 0.93) | 0.004 |
| Total antenatal care visits | 0.62 (0.41 to 0.83) | 0.86 (0.72 to 0.99) | 0.050 |
| Antenatal care syphilis testing | 0.08 (−0.02 to 0.18) | 0.59 (0.42 to 0.76) | <0.001 |
| Facility deliveries | 0.48 (0.26 to 0.70) | 0.82 (0.69 to 0.94) | 0.019 |
| Use of partograph to monitor labour and delivery | 0.24 (0.03 to 0.45) | 0.83 (0.77 to 0.89) | <0.001 |
| Total postnatal care visits | 0.15 (0.02 to 0.29) | 0.58 (0.38 to 0.78) | 0.001 |
| Early postpartum-postnatal care within 3 days of birth | 0.21 (0.01 to 0.41) | 0.55 (0.31 to 0.80) | 0.020 |

ICC values range from 0 to 1, with values approaching 1 representing greater agreement.
ICC, intraclass correlation coefficient.

## External consistency: agreement between facility summary reports in DHIS2 and household surveys

Figure 2 summarises external consistency, which is the agreement between facility-based routine data in DHIS2 compared with household-level surveys in the enumeration areas of these facilities. Other than the indicator for early postpartum-postnatal care, there was no agreement nor any consistent pattern of agreement between facility-based routine data and the household surveys. DHIS2 data underestimated compared with the household survey for at least one dose of intermittent preventative therapy for malaria in pregnancy and at least one dose of tetanus toxoid. DHIS2 data overestimated compared with the household survey for baby weighed at birth, oral polio vaccine given at birth and BCG given during postnatal period. For antenatal care anaemia testing, facility-based

estimates were within the household survey estimate CI but overestimated compared with the household survey during the intervention period.

## DISCUSSION

Facility-based routine data are an important source for monitoring, performance management and planning.[9] Our study found that an integrated district-focused data quality intervention—which included regular self-assessment of data quality, peer-review and feedback, learning workshops, work planning for improvement, monthly data quality reports, and ongoing support through social media—was associated with improvements across most WHO data quality metrics. There were differences in data quality improvement by data element type.[50] Data related to content of care or the provision of commodities, such as syphilis testing and intermittent preventative therapy for malaria, improved more across data quality metrics compared with contact indicators. Contact indicators had relatively higher data quality metrics before the intervention, such as first antenatal care visits and facility deliveries. Contact indicators such as first antenatal care visits, fourth antenatal care visits and facility deliveries are generally well-defined events to document, have been key denominators for local programme planning, and have been prioritised for monitoring progress in previous global initiatives including the Millennium Development Goals and Countdown to 2015.[50–52]

This was an integrated data quality intervention designed to facilitate existing state-level and district-level data quality checking responsibilities and emphasise the partnership between the monitoring and evaluation officers and the maternal, newborn and child health programme coordinators to expand local access to the DHIS2 data, use the data and problem solve.[9] Incorporating regular data quality reviews and feedback within supervision is an acknowledged low-cost and effective activity for improving completeness and consistency.[6 8 17 18 53] Leveraging existing staff and meeting schedules as well as adding structure to existing data-related responsibilities further

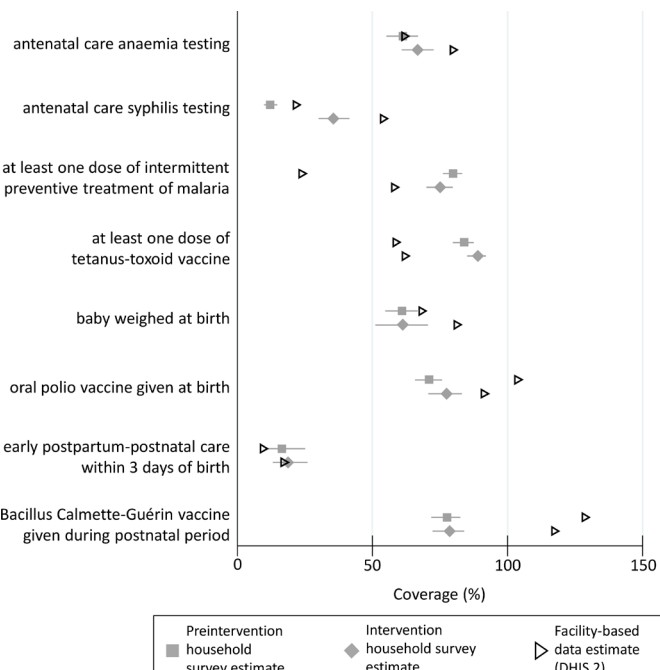

**Figure 2** External consistency: comparison of household-level survey and facility summary reports in District Health Information Software 2 (n=97 facilities).

promotes the feasibility of this intervention in similar settings. The evolution of the intervention through the work planning sessions prompted local solutions defined by the participants as feasible and within their resources to implement. In particular, the participants' decision to revitalise the data validation committees during the first workshop engaged the facilities early on to ensure the facility's register counts matched the facility's monthly report aggregate value. This early engagement with facilities could have contributed to the increased ICCs for accuracy of facility reporting observed during the intervention period (as shown in table 3). A formative phase of the intervention might have captured activities such as the data validation committee as a predefined intervention activity. Including a formative phase should be a consideration for future implementation.

Our findings aligned with previous studies reporting improvements in completeness, timeliness and accuracy of facility reporting after intervention. However, this study's relative gains may reflect the scale of working through 11 districts with 492 primary health facilities.[4 6 8 35 36] A data quality intervention in KwaZulu-Natal province, South Africa, which included trainings, monthly data meetings and external data quality audits across 78 facilities improved completeness of six data elements from 26% to 64% and the agreement between facility records and reports (data accuracy) from 37% to 65%.[6] A province-wide data quality intervention in Sofala, Mozambique, for 26 facilities included regular district-level review meetings for health workers and managers, data dashboards for tracking trends and rankings, human resource optimisation models and equipment purchase and maintenance. The summary measure used to evaluate data quality improvement, concordance, improved from 56% to 88% during the intervention period.[4] The introduction of an electronic medical record to support data quality improvement in 27 facilities across Kenya recorded a decline in missing data from 31% to 13% for 24 data elements, with a mean concordance score increasing across facilities by 1.79 (95% CI 0.25 to 3.33).[36]

While our findings align with previous studies for increased completeness, timeliness and accuracy, our study reviewed additional WHO metrics to give a more comprehensive picture of the dimensions of data quality. This study also provided an opportunity to reflect on the relative usefulness of assessing all WHO data quality metrics to understand the quality of routine data in a given context. Sharp increases in service uptake due to health campaigns or targeted health projects make the assessment of moderate outliers and consistency over time less insightful about data quality, especially in the context of urgent efforts towards achieving universal health coverage. Our study found that, other than early postpartum-postnatal care within 3 days of birth, there were no instances of agreement between the facility-based routine data and external household surveys. However, an emerging body of criterion validity studies have demonstrated mixed results in the ability of women to recall facility-based pregnancy-related and childbirth-related events in household surveys.[54–58] More research is needed on how to reconcile health facility and household survey data, while also reconsidering the emphasis on the household surveys as the reference standard.

The data quality metric to assess the consistency between related data elements provided useful insights in addition to the completeness and accuracy regularly reported in the literature. Assessing the data relationship for a service provision compared with a contact indicator (eg, antenatal care anaemia testing compared with antenatal care first visits) allows for discussion on whether observed discrepancies are due to low service uptake or poor reporting, an important consideration given the emphasis on improving quality of care and understanding effective coverage.[59] Further, assessing data relationships that require concurrent tallying of services/information across data sources (eg, facility attendance=inpatient+outpatient; normal delivery+caesarean delivery+assisted delivery=live births+still births), provides useful insight about whether a facility is paying attention to the internal consistency of their data within and across facility documentation. Focus on accuracy of facility reporting, a more common metric assessed in the peer-reviewed literature (referred to as data accuracy or concordance), is an important data quality metric as subnational, national and global-level monitoring cannot take place effectively without the confidence that the facilities have summarised and tallied the data as intended. However, this focus on the accuracy of facility reporting up through the different levels of the health system do not require that these data be internally consistent with other data.

Our study had limitations. Without a concurrent comparison group, our before-and-after analyses cannot eliminate the effects of concurrent events and activities on data quality metrics. It is possible that other activities contributed to the observed data quality improvements. Given the high burden of maternal and neonatal mortality, the Gombe State Primary Health Care Development Agency spearheaded an initiative to improve maternal and neonatal services with the aim of having one fully functional primary health facility in each of its 114 wards. During the intervention period, 57 facilities (12% of the 492 maternity facilities) received support including facility-level quality improvement support as well as community-based outreach and education to increase uptake of services. Facility-level activities included support on data quality to monitor trends in services provided and the provision of computers and facility registers. Additionally, similar to other data quality assessments, we did not validate the data through direct clinical observations[4 6 10 12 32 60 61] nor did we compare the paper-based monthly summary reports to their electronic versions in DHIS2.[6 12 37 62–64] Despite close attention to quality control, the facility-level and household-level surveys might still be susceptible to errors in data recording, including incorrectly tallying the number of events in the original facility registers for comparison with data in DHIS2.

Improving the quality of routine facility data is essential for local and national evidence-based monitoring of universal health coverage. We found that an integrated district-focused data quality intervention was associated with increases across most WHO data quality metrics for routine facility-based data. Future initiatives should aim to incorporate national-level and higher subnational-levels of the health system to determine scalability and sustainability of integrated data quality interventions in the long-term.

**Acknowledgements** The authors wish to acknowledge the leadership of the Gombe State Primary Health Care Development Agency throughout the development and implementation of the data quality intervention. We also appreciate the cooperation of and partnership with the LGA M&E officers, MNCH coordinators, and Gombe State implementing partners to improve the quality of routine maternal and newborn health data in Gombe State. We are grateful to the women who participated in the household surveys and to the team who conducted the facility and household data collection. Finally, we are grateful to the facility teams responsible for providing and documenting their care for women and newborns.

**Contributors** AAB and TM conceived and designed the study. AAB and EA designed the analyses. AAB carried out the analyses and composed the initial draft. JS, TM, EA, NU, AA, HF reviewed the early drafts. All authors approved the final draft.

**Funding** This work was supported by the Bill & Melinda Gates Foundation (OPP1149259). The funder of this study had no role in the study's design or conduct, data collection, analysis or interpretation of results, writing of the paper or decision to submit for publication.

**Competing interests** HF and AA are members of the Gombe State Primary Health Care Development Agency. The authors declare no other competing interests.

**Patient consent for publication** Not required.

**Ethics approval** Study approvals were obtained from the London School of Hygiene & Tropical Medicine (reference 14091) and the Health Research Ethics Committees for Nigeria (reference NHREC/01/01/2007) and Gombe State (reference ADM/S/658/Vol. II/66).

**Provenance and peer review** Not commissioned; externally peer reviewed.

**Data availability statement** Data are available in a public, open access repository. Data for this study are available from the London School of Hygiene & Tropical Medicine public repository: http://datacompass.lshtm.ac.uk/229/.

**ORCID iD**
Antoinette Alas Bhattacharya http://orcid.org/0000-0001-5400-9383

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
