## [Reviewer comments · BMJ Open]

ARTICLE DETAILS

TITLE (PROVISIONAL)	Improving the quality of routine maternal and newborn data captured in primary health facilities in Gombe State, northeastern Nigeria: a before-and-after study
AUTHORS	Bhattacharya, Antoinette Alas; Allen, Elizabeth; Umar, Nasir; Audu, Ahmed; Felix, Habila; Schellenberg, Joanna; Marchant, Tanya

VERSION 1 – REVIEW

REVIEWER	Maria Ospina University of Alberta (Canada)
REVIEW RETURNED	17-Mar-2020

GENERAL COMMENTS	The manuscript by Bhattacharya et al describes the development of an integrated district-focused data quality intervention in 11 local government areas in northeastern Nigeria and quantified changes in data quality metrics before and after implementing the intervention. The intervention included actions such as self-assessment of data quality, peer-review and feedback, workshops, workplanning and social media support. Nine metrics were selected for evaluation, pertaining to WHO main quality dimensions of 1) completeness and timeliness, 2) internal consistency, and 3) external consistency. These metrics were assessed for 14 maternal and newborn health data elements in a before-and after study. The work presented in this manuscript is very important to inform policy makers on maintaining high-quality routine information systems to inform performance and optimization of service delivery while monitoring/addressing inequalities in the provision of services and health outcomes. Overall, the manuscript is well-written. The manuscript describes a quality improvement project rather than research, it has the potential to make an important contribution to the literature on patient care and process indicators. I am not sure about how the work presented aligns with BMJ Open scope of work, as the manuscript seems to fit better with a health care quality improvement journal. I would like to provide some detailed comments for the authors' consideration to improve the manuscript. Thank you for the opportunity to review this manuscript. - The Introduction provides a succinct but very complete background to the study. It highlights the knowledge gaps regarding the quantification of changes in quality metrics resulting from interventions to improve these processes. I suggest the authors provide some references and expand the description of
--

	the evidence on the effectiveness of interventions to improve the quality of routine data collection. The authors cite a few of these interventions (e.g., training sessions, workshops, surveys, etc) but nothing is said about the level of evidence to endorse these interventions.  - The description of the study setting and data quality intervention is very comprehensive. Please provide an evidence-based justification of why the proposed actions for data quality intervention were selected. - Please clarify why only 3 out of 4 WHO routine data quality dimensions were evaluated in this study. Why external comparisons were not incorporated in this evaluation. - Please clarify the description of the statistical methods for the evaluation of completeness and timeliness. Are these evaluated by comparing changes in proportions?, which statistical methods were chosen. What about timeliness? - Please clarify how potential confounders were addressed in the analysis.
--	--

REVIEWER	Jessica Chiliza Centre for Rural Health, University of KwaZulu-Natal, South Africa
REVIEW RETURNED	19-Apr-2020

GENERAL COMMENTS	The word workplanning is two words: work planning.  - Capitalize title of each section (line 30) - Pg. 6 I don't understand this sentence.(lines 40-46): "It was designed as a facilitative layer to existing LGA level supervision responsibilities, leveraging scheduling of ongoing activities to minimize cost, adding job aids and defining performance standards to provide structure to data quality checking duties and to target feedback to facilities." - Pg. 8 Capitalize the names of the departments (line 17-22) - Pg. 7 Line 45, instead of "dormant groups" I would say "strategies" or something like that - Please clarify what the enumeration areas were. Are they same as catchment areas (pg 9 line 37)? Be consistent with language. Pg. 9 line 52 add "to" be interviewed. Pg. 13 line 40-43: missing words... DHIS2 data underestimated? Discussion: It would be important to note how feasible this intervention would be in a real world setting. Line 17-27: Why do you think the content of care indicators improved more than contact indicators?
---

REVIEWER	Omar Yaxmehen Bello-Chavolla Instituto Nacional de Geriatria
REVIEW RETURNED	08-Jun-2020

GENERAL COMMENTS	Interesting paper which aims to evaluate quantitatively the impact of a data intervention on improving data quality and accuracy of facility-based routine data. Overall, the paper are well-written and the results supports the authors hypothesis. I have performed a statistical review of this manuscript, which I find strong provided the authors address some concerns:  1. Authors present comparison of before and after percent changes in report timeliness and availability and present p-values. Which statistical test was used to compute them? Was it a specific test or a confidence interval test? McNemar's chi-squared test
--

	seems appropriate but is not reported in the methods section, please specify. 2. Whilst confidence interval comparison of ICC in independent samples is feasible, comparison in correlated samples (which, in this case, is considered within the structure of a before-after design) requires consideration of a dependence structure in the estimators. These methods have been developed recently and authors are encourage to either consider its estimation or to discuss the potential pitfalls of their approach (https://arxiv.org/abs/1809.09740). 3 "During the preintervention period, one of the 10 relationships reflected high consistency: iron-folic acid supplementation. During the implementation period, five of the 10 relationships reflected improved consistency". These claims support the need to consider a correlated structure since baseline adjustments would readily be considered, the used of paired tests must be clarified throughout the paper. 4. Internal consistency can also be assessed using Chronbach's alpha coefficient. this is complementary to ICC and gives a holistic overview metric of internal consistency. Have the authors considered calculating this metric? (eg. doi:10.1080/02813430802294803, doi: 10.1007/s11165-016-9602-2).
--	--

VERSION 1 – AUTHOR RESPONSE

Reviewer-1:

Reviewers' Comments	Authors' Response
The Introduction provides a succinct but very complete background to the study. It highlights the knowledge gaps regarding the quantification of changes in quality metrics resulting from interventions to improve these processes. I suggest the authors provide some references and expand the description of the evidence on the effectiveness of interventions to improve the quality of routine data collection. The authors cite a few of these interventions (e.g., training sessions, workshops, surveys, etc) but nothing is said about the level of evidence to endorse these interventions.	Thank you for this feedback. As suggested, we have expanded the description of the evidence on interventions to improve the quality of routine data. Please see the suggested revisions, starting with paragraph-3 of the Introduction.
The description of the study setting and data quality intervention is very comprehensive. Please provide an evidence-based justification of why the proposed actions for data quality intervention were selected.	Thank you for this feedback. As suggested, we have expanded the description of how the data quality intervention activities were selected. Please see the suggested revisions, starting with paragraph-1 under the 'Data quality intervention' sub-section in the Methods section.
Please clarify why only 3 out of 4 WHO routine data quality dimensions were evaluated in this study. Why external comparisons were not incorporated in this evaluation.	Thank you for this feedback and observation. The last available census data for Nigeria and its states are from 2006; the most recent census data are not yet publicly available. There are

	slightly different projections for the population of Nigeria by the National Bureau of Statistics and by the United Nations World Population Prospects run by the Department of Economic and Social Affairs. However, for Gombe State-level projections, the United Nations and multiple health programs use the National Bureau of Statistics projection data. As the purpose of data quality dimension 'external comparisons' is to compare data derived from two different sources, the 'external comparisons' dimension did not seem suitable to compare two identical numbers (originating from the same source), though they are being used by different agencies and programs.
Please clarify the description of the statistical methods for the evaluation of completeness and timeliness. Are these evaluated by comparing changes in proportions?, which statistical methods were chosen. What about timeliness?	Thank you for this feedback. We have expanded our description of the statistical methods as well as specific explanations on evaluating completeness and timeliness, as requested. Please see our suggested revisions, starting on paragraph-4 under the 'Data analyses and data sources' sub-section of the Methods.
Please clarify how potential confounders were addressed in the analysis.	Thank you for this feedback. We have also included this clarification, starting on paragraph-4 under the 'Data analyses and data sources' sub-section of the Methods.

Reviewer-2:

Reviewers' Comments	Authors' Response
The word workplanning is two words: work planning.	Thank you for this feedback. We have amended the spelling, as suggested.
Capitalize title of each section (line 30)	Thank you for this feedback. We were unable to find sections which were not capitalized, as our document version appears to have each section capitalized. We are happy to make any further amendments if the reviewer could provide more details.
Pg. 6 I don't understand this sentence.(lines 40-46): "It was designed as a facilitative layer to existing LGA level supervision responsibilities, leveraging scheduling of ongoing activities to minimize cost, adding job aids and defining performance standards to provide structure to data quality checking duties and to target feedback to facilities."	Thank you for this feedback. We have amended the original sentence as follows: "The intervention was designed to facilitate existing LGA level supervision responsibilities. It leveraged scheduling of ongoing activities to minimize cost, added job aids, and defined performance standards to provide structure to existing data quality checking duties and to target feedback to facilities."

Pg. 8 Capitalize the names of the departments (line 17-22)	Thank you for this feedback. As suggested, we have capitalized the department name.
Pg. 7 Line 45, instead of “dormant groups” I would say “strategies” or something like that	Thank you for this feedback. As suggested, we have amended this to “dormant strategies”.
Please clarify what the enumeration areas were. Are they same as catchment areas (pg 9 line 37)? Be consistent with language.	Thank you for this feedback. As suggested, we have changed “catchment” areas to “enumeration” areas in all three instances used in the manuscript to avoid confusion.
Pg. 9 line 52 add “to” be interviewed.	Thank you for this feedback. We have made the edit, as suggested.
Pg. 13 line 40-43: missing words... DHIS2 data underestimated?	Thank you for this feedback. As suggested, we have changed this to “DHIS2 data”.
Discussion: It would be important to note how feasible this intervention would be in a real world setting.	Thank you for this feedback. As suggested, we have noted the feasibility of this intervention in paragraph-2 in the Discussion section.
Discussion: Line 17-27: Why do you think the content of care indicators improved more than contact indicators?	Thank you for this question and the opportunity to elaborate on the data quality improvement by indicator type. Please see the end of paragraph-1 in the Discussion where we have noted the following: “Contact indicators such as first antenatal care visits, fourth antenatal care visits, and facility deliveries are generally well-defined events to document, have been key denominators for local program planning, and have been prioritized for monitoring progress in previous global initiatives including the Millennium Development Goals and Countdown to 2015.”

Reviewer-3:

Reviewers' Comments	Authors' Response
Authors present comparison of before and after percent changes in report timeliness and availability and present p-values. Which statistical test was used to compute them? Was it a specific test or a confidence interval test? McNemar's chi-squared test seems appropriate but is not reported in the methods section, please specify.	Thank you for this feedback. As suggested, we added language to the ‘Data analyses and data sources’ sub-section within Methods to clarify the tests used in assessing the data quality metrics. To examine changes in the completeness and timeliness metrics, we used mixed models to account for the clustering of facility measurements within facilities and the clustering of facilities within districts. P-values for changes in these data quality metrics were computed from the Wald test within the mixed models.
Whilst confidence interval comparison of ICC in independent samples is feasible, comparison in correlated samples (which, in this case, is considered within the structure of a before-after design) requires consideration of a dependence structure in the estimators. These methods have been developed recently and authors are encourage to either consider its estimation or to	First, we would like to appreciate all of the thoughtful feedback provided by Reviewer-3 during the statistical review. It provided an opportunity for our team to speak again at length and carefully consider his suggestions in light of questions we intended to answer through the analyses.

discuss the potential pitfalls of their approach (https://arxiv.org/abs/1809.09740).	We address the second and third comments from Reviewer-3 together:
"During the preintervention period, one of the 10 relationships reflected high consistency: iron-folic acid supplementation. During the implementation period, five of the 10 relationships reflected improved consistency". These claims support the need to consider a correlated structure since baseline adjustments would readily be considered, the used of paired tests must be clarified throughout the paper.	The study aims to evaluate the quality of routine data for the pre-intervention and intervention periods of a district-focused intervention in Gombe State, using a before-and-after study design. The intervention worked through district-level staff to improve data quality across 492 Gombe State facilities. In the manuscript originally submitted, ICCs were generated for each data relationship (e.g., normal + assisted + caesarean deliveries = live births + still births). Given facility personnel changes and staff attrition over the 42-month study period, we assessed overall Gombe State changes in data quality metrics and did not match facilities to determine within-facility improvements. This would be to understand if the intervention had an overall population-level change in data quality, rather than facility-level. We conservatively looked at the confidence intervals for the pre-intervention and intervention periods; if the confidence intervals did not overlap, then we concluded that agreement had improved. Based on Reviewer-3's feedback, we discussed and re-adjusted our analyses to match at the LGA-level (district-level equivalent), given that the intervention was delivered to the LGA M&E officer and LGA MNCH coordinators. This staffing was relatively stable and an ICC could be generated for each LGA. We generated an ICC per LGA, following the same approach as before where we partitioned the variance into between-facility variation within the LGA, between-measures variation to understand the level of agreement between the data relationship, and the error (remaining variability). We then conducted a paired t-test, which paired ICCs at the LGA-level for the pre-intervention and intervention periods. The paired test was a more powerful test, as suggested by Reviewer-3. Compared to the originally submitted results, there was a notable reduction in the standard error for two data relationships for the data quality metric 'consistency between related data': (i) antenatal care anemia testing and (ii) antenatal care syphilis testing. The other data relationships remain

	unchanged from the original submission; their paired and unpaired tests were similar. Our updated analyses are reflected in Tables 2 and 3. We converted the original Figures 2 and 3 into Tables 2 and 3, respectively, so that the data are more visible to the reader. We hope Reviewer-3 might find this approach acceptable. We have noted the edits in the Methods and the Results sections.
Internal consistency can also be assessed using Chronbach's alpha coefficient. this is complementary to ICC and gives a holistic overview metric of internal consistency. Have the authors considered calculating this metric? (eg. doi:10.1080/02813430802294803, doi: 10.1007/s11165-016-9602-2).	Thank you for this feedback. Again, we would like to appreciate Reviewer-3's constructive comments for improving this manuscript. This work is based on the data quality dimensions and metrics guidance, as outlined in the WHO data quality review toolkit for facility data. For the two data quality metrics where ICC is being used – (i) consistency between related data elements and (ii) accuracy of facility reporting – the objective was to assess the extent of absolute agreement of the data values. Our initial analyses considered calculating the agreement as originally proposed by the WHO toolkit: a verification factor which is the ratio of value#1 to value#2. Given the focus on absolute agreement, we also analysed using Bland-Altman's limits of agreement and the ICC. Ultimately, we found that the ICC was the most efficient method for communicating the agreement results for (i) 10 data relationships for assessing consistency between related data and (ii) 7 data relationships for assessing accuracy of facility reporting. In a follow-on manuscript where we consider developing a score for data quality based on the WHO data quality dimensions for routine facility data, we intend to use Cronbach's alpha to examine the individual dimensions of data quality for constructing the composite score.

VERSION 2 – REVIEW

REVIEWER	Omar Yaxmehen Bello-Chavolla Instituto Nacional de Geriatria, Mexico
REVIEW RETURNED	29-Jul-2020

GENERAL COMMENTS	The authors have adequately addressed all concerns. The statistical methods have improved upon revision and the added transparency by presenting the estimates in tables rather than figures is much appreciated. Thanks to the authors for their thorough and considerate revision.
--

VERSION 2 – AUTHOR RESPONSE

We are pleased that the BMJ Open editors and reviewers have recommended publication of our manuscript. We especially thank Dr Omar Yaxmehen Bello-Chavolla (Review #3) for his careful review and publication recommendation of the revised manuscript.

The remaining revision is one minor revision, as given by the BMJ Open editors. We have revised the manuscript title according to the preferred format of the journal. The revised title is: "Improving the quality of routine maternal and newborn data captured in primary health facilities in Gombe State, northeastern Nigeria: a before-and-after study". We hope this revised title is satisfactory.

We are happy to make any further revisions requested.